# Alterations in the Nervous System and Gut Microbiota after *β*-Hemolytic Streptococcus Group A Infection—Characteristics and Diagnostic Criteria of PANDAS Recognition

**DOI:** 10.3390/ijms21041476

**Published:** 2020-02-21

**Authors:** Jacek Baj, Ryszard Sitarz, Alicja Forma, Katarzyna Wróblewska, Hanna Karakuła-Juchnowicz

**Affiliations:** 1Chair and Department of Anatomy, Medical University of Lublin, 20-090 Lublin, Poland; 2Department of Forensic Medicine, Medical University of Lublin, 20-090 Lublin, Poland; e.sitarz@hotmail.com (R.S.); aforma@onet.pl (A.F.); 3North London Forensic Service, Chase Farm Hospital, 127 The Ridgeway, Enfield, Middlesex EN2 8JL, UK; k.wroblewska@gmail.com; 4Chair and 1st Department of Psychiatry, Psychotherapy and Early Intervention, Medical University of Lublin, Gluska Street 1, 20-439 Lublin, Poland; karakula.hanna@gmail.com; 5Department of Clinical Neuropsychiatry, Medical University of Lublin, Gluska Street 1, 20-439 Lublin, Poland

**Keywords:** pediatric autoimmune neuropsychiatric disorders associated with streptococcal infections, PANDAS, β-hemolytic *Streptococcus* group A, PANS, pediatric acute-onset neuropsychiatric syndrome, nervous system, gut microbiota, psychiatry, obsessive-compulsive disorder, diagnostic criteria

## Abstract

The objective of this paper is to review and summarize conclusions from the available literature regarding Pediatric Autoimmune Neuropsychiatric Disorders Associated with Streptococcal Infections (PANDAS). The authors have independently reviewed articles from 1977 onwards, primarily focusing on the etiopathology, symptoms, differentiation between similar psychiatric conditions, immunological reactions, alterations in the nervous system and gut microbiota, genetics, and the available treatment for PANDAS. Recent research indicates that PANDAS patients show noticeable alterations within the structures of the central nervous system, including caudate, putamen, globus pallidus, and striatum, as well as bilateral and lentiform nuclei. Likewise, the presence of autoantibodies that interact with basal ganglia was observed in PANDAS patients. Several studies also suggest a relationship between the presence of obsessive-compulsive disorders like PANDAS and alterations to the gut microbiota. Further, genetic predispositions—including variations in the *MBL* gene and TNF-α—seem to be relevant regarding PANDAS syndrome. Even though the literature is still scarce, the authors have attempted to provide a thorough insight into the PANDAS syndrome, bearing in mind the diagnostic difficulties of this condition.

## 1. From Acute Pharyngitis to Rheumatic Fever

Acute pharyngitis and tonsillitis may be induced by various bacterial and viral organisms, even though β-hemolytic Streptococci group A remains the most common causation so far [1,2,3]. *Streptococcus pyogenes* (β-hemolytic *Streptococcus* group A) (GAS) is responsible for the majority of bacterial infections in children [4]. The highest infection rate is estimated to occur during late autumn, winter, and early spring. An infected human constitutes a reservoir of pathogenic bacteria and a potential source of infection. GAS infection can be spread by airborne transmission or through direct contact with an infected individual. Streptococcal pharyngitis is characterized by the following symptoms: rapid and acute onset, sore throat, pain during swallowing, headache, nausea, vomiting, fever, intensely red-coloured oral mucosa with swelling, as well as painful and enlarged anterior cervical lymph nodes [5]. Coughing and rhinitis are not common symptoms of streptococcal infection. Post-infectious streptococcal complications mainly include peritonsillar abscesses, purulent otitis media, or paranasal sinusitis [6]. Further, streptococcal toxic shock syndrome, post-streptococcal acute glomerulonephritis, acute rheumatic fever, rheumatic heart disease, or post-streptococcal autoimmune neuropsychiatric disorders (so-called Pediatric Autoimmune Neuropsychiatric Disorders Associated with Streptococcal Infections, or PANDAS) may be induced by autoimmune responses [7]. There are also incidents of non-streptococcal infections by the Bornavirus or *Mycoplasma pneumoniae* that are implicated in the onset of symptoms similar to those in PANDAS [8,9,10]. Exacerbations of PANDAS symptoms are observed in 30% of cases due to GAS infections; 20% are because of non-streptococcal illnesses and approximately half of the cases are because of a non-identified factor [11].

## 2. Rheumatic Fever and OCD

Rheumatic fever is an acute autoimmune disease constituting a relatively late complication of GAS infection [12,13]. The Jones criteria for rheumatic fever include major criteria (carditis, arthritis, chorea, erythema marginatum, subcutaneous nodules) and minor criteria (polyarthralgia, fever, prolonged PR interval, erythrocyte sedimentation rate (ESR) ≥60 mm, C-reactive protein (CRP) ≥3.0 mg/dL) [14]. The symptoms of rheumatic fever appear usually two-to-three weeks after pharyngitis and include the inflammation of large joints, myocarditis, marginal erythema, subcutaneous nodules, or even incidents of Sydenham’s chorea [15,16]. The number of acute rheumatic fever incidents occurring as a side effect of streptococcal infection equals 5%–6% and depends on the susceptibility of an infected individual, the duration of GAS exposure, and the type of treatment therapy (or even lack of appropriate treatment) [17]. The pathomechanism of rheumatic fever involves immunological reactions directed towards specific epitopes with a structure similar to the proteins present in the myocardium, heart valves, synovium, and skin, as well as in the hypothalamus and caudate nucleus [18,19].

Approximately 2%–4% of children with rheumatic fever are prone to developing OCD, which is also associated with the progressive damage of basal ganglia [20,21]. This phenomenon shows a male predominance and may also manifest as tics, Tourette’s syndrome, or attention-deficit/hyperactivity disorder (ADHD) [22]. Obsessive-compulsive disorders (OCD) might also accompany Sydenham’s chorea. Sydenham’s chorea manifests in the presence of unilateral involuntary movements—mainly of facial and limb muscles—general weakness, and emotional instability [23]. The psychiatric symptoms may appear insidiously and include emotional instability, irritability, anxiety, and inappropriate behavior in general. Additionally, affected children may present progressively worse results in school. In both PANDAS and Sydenham’s chorea, some new evidence supports the concept of autoantibody mimicry mechanisms [24,25]. The antibodies, which induce either rheumatic fever or PANDAS, are cross-reactive with the *N*-acetyl-βD-glucosamine dominant epitope specifically [26].

## 3. PANDAS and OCD—Where Is the Line?

The first description of PANDAS was presented in 1998 by Swedo et al. based on the 50 cases she reported, defining it as the sudden beginning of obsessive-compulsive or tic disorders’ symptoms due to the complications of GAS infection [27]. The rapid onset of such symptoms hinders the differentiation between OCD and PANDAS [28]. PANDAS involves various diagnostic implications since the profile of symptoms seems to be similar to other streptococcal-associated conditions, like OCD, tics, or motor hyperactivity [29,30]. Further, the majority of PANDAS cases described by Swedo included the presence of insignificant piano-playing choreiform movements of fingers and toes, which may be overlooked [31]. Symptoms similar to PANDAS may present not only in children but also among adults as a side effect of post-streptococcal disease [32]. Besides sore throats due to the GAS infection, there are cases of streptococcal skin infections, which may also be associated with the onset of PANDAS [33]. The OCD symptoms constitute the foundational criteria for PANDAS recognition, as well. It was estimated that the post-streptococcal autoimmunity mechanism is responsible for up to 10% of cases of childhood OCD [19]. To resolve the diagnostic issue, it was hypothesized that PANDAS constitutes a subtype of children’s OCD, with the etiology of GAS infection. The legitimacy of PANDAS as a subgroup of OCD was supported by the study performed by Jaspers-Fayer et al. in 2017 [34]. 

### PANDAS Symptoms

PANDAS mostly occurs in children or adolescents and its prevalence is significantly higher in males [35,36,37,38,39,40,41]. PANDAS is characterized by impairments within basal ganglia, which are responsible for the switching of both motor and mental behaviours, resulting in novel behaviour [42]. PANDAS children present characteristic symptoms such as tics, hyperactivity, urinary urgency, impulsivity, anxiety, impulsiveness, eating disorders, and a significant decline in school performance along with the deterioration of handwriting [43,44]. Usually, children with PANDAS show significant and rapid (between 24 and 72 h) changes in behavior, shifting from high-functioning and well-adjusted to psychotic symptoms which may disrupt children’s functioning [45]. On average, anxiety and other mood symptoms are more severe compared to somatic and functional symptoms like stomach, muscle, or joint pain [46].

There is a positive correlation between upper respiratory infections, the sudden onset of OCD, and PANDAS symptoms [47]. Nonetheless, other risk factors including the age of onset, neurologic impairments, family history of either autoimmune disorders or rheumatic fever, the presence of comorbidities or the severity of OCD symptoms seem not to be correlated with the presence of upper respiratory infections [47,48,49].

According to Leon et al. (2017), a complete PANDAS remission is, on average, 3.3 years [11]. Moreover, it was observed that the majority of children (approximately 72%) showed at least one exacerbation of PANDAS symptoms throughout the period of gradual remission. Among PANDAS cohorts, psychiatric comorbidities included ADHD (40%), oppositional defiant disorder (40%), and depression (36%). Less common were enuresis (20%) or dysthymia (12%) [50]. ADHD-like behavioral changes are common side effects of concurrent or prior streptococcal infections [51]. Male predominance was observed in ADHD and PANDAS; males are also more prone to recurrent GAS infections [52]. PANDAS symptoms may be similar to those which are present in other psychiatric disorders, including OCD, Sydenham’s chorea, Tourette’s syndrome, or Tumor Necrosis Factor Receptor Associated Periodic Syndrome (TRAPS) [53].

## 4. PANDAS Diagnostic Guidelines

Guidelines regarding diagnostic criteria for PANDAS were improved in 2017, as it includes:*Presence of OCD and/or tics, mainly multiple ones, complex or not observable in other disorders*
*Specific period of childhood and age (symptoms of PANDAS are more common between the age of 3 years old and period of puberty)*
*Acute onset and episodic changes in behavior*
*Association with infection of Streptococci group A*
*Coexisting neurological impairments* [54,55].

### 4.1. The Presence of OCD and/or Tics

Obsessions, compulsions and/or tics should be severe to meet either OCD or tics criteria and distort patients functioning abilities comparing to the state before the disease.

### 4.2. Specific Period of Childhood and Age

Symptoms are most frequently observed between the age of 3 and puberty. The early onset of PANDAS is associated with the time of the highest rate of GAS exposition (early childhood). The onset of PANDAS after puberty is possible, nevertheless, it forms only in a minor percentage of cases.

### 4.3. Acute Onset and Episodic Changes in Behavior

Clinical outcome is characterized by the rapid and dramatic onset of OCD, plus tics. The beginning of specific symptoms may usually be assigned to a particular day or week, which confirms that the onset is relatively rapid and obvious. The prolonged duration of the streptococcal infection could worsen the clinical outcome. Nevertheless, to comply with this criterion, deterioration of tics must be of such severity as to restrain the patient. The varying severity of symptoms depends on the episode of the disease. Moreover, because children are potential carriers of various range of pathogens, the symptomatic period might slightly be prolonged depending on the patient.

### 4.4. Association with Infection of Streptococci Group A

The exacerbation of symptoms must be associated with the infection of group A Streptococci, which needs to be confirmed with the positive swab from the throat and/or increased titers of antistreptolysin-O (ASO) or anti-DNase B [56,57]. In the case of PANDAS, GAS infection usually occurs without macroscopic changes in the throat and symptoms of pharyngitis. It is recommended to provide a 24–48 h agar culture to confirm the colonization of GAS [58]. Positive findings may indicate the carrier state of GAS in a patient without any specific immunological reactions. With regards to PANDAS, it was detected that neuropsychiatric symptoms are associated with the presence of immunological reactions, and can be eliminated after the removal of antibodies. The increase of the titers of either ASO or anti-DNase B can be used in the detection of previous infections of GAS.

Therefore, the usage of this diagnostic method could slightly be misleading due to a number of reasons:Titers of either ASO or anti-DNase B can remain at a high level even for many months after GAS infection, which might present as a false positive result.Approximately 40% of children with GAS infection do not present the increased levels of ASO or anti-DNase B which is a false negative result.Time constitutes an essential critical factor in the determination of the increased levels (approximately 2–4 times higher) of ASO, and anti-DNase B after one-four and six-eight weeks correspondingly.When a child with specific symptoms presents negative results from a throat swab, the levels of basic anti-streptococcal antibodies should be provided.

Studies have shown that after initial infection, disease exacerbations could be associated with factors other than GAS, such as other bacterial or viral infections, or internal stimuli like stress [59].

### 4.5. Coexisting Neurological Impairments

During the exacerbation of symptoms, patients show inappropriate results from the neurologic examination. The most common impairments include hyperactivity or involuntary movements (tics or chorea). It is essential to differentiate between PANDAS and Sydenham’s chorea since the last one is the common symptom of rheumatic fever and present different, separate characteristics and treatment strategies. Approximately 30% of patients with either OCD or PANDAS present choreiform movements that are similar to those in Sydenham’s chorea, which eventually might raise diagnostic issues (Table 1) [22].

Etiologically in both PANDAS and Sydenham’s chorea present as a side effect of GAS infection, nevertheless, besides the presence of tics, which is common for both diseases, Sydenham’s chorea appears with more severe obsessive-compulsive syndromes and definite chorea with the presence of hypotonia [93,94]. Moreover, Sydenham’s chorea usually presents with complete remissions and a duration less than one year, whereas the PANDAS course is likely to be more chronic [95]. Furthermore, PANDAS found to be more prevalent in men, whereas Sydenham’s chorea in women. Regarding the streptococcal M-protein subtypes—PANDAS is associated with M1, M3, M11, M12, and M13, whereas Sydenham’s chorea—M5, M6, and M19 [96].

Diagnostic criteria for PANDAS require the presence of OCD or tics in general, nonetheless, neuropsychiatric symptoms are also commonly present among patients. They appear simultaneously with OCD or tics, and similarly present sudden onset and exacerbation. It has been shown that the presence of somatic disturbances like higher frequency of urination, pupil dilatation, and insomnia aids in the differentiation between PANDAS and OCD, or Tourette’s syndrome [97]. The affected children may present signs of inattention, impulsiveness, emotional lability for days, weeks, or even months after remission of OCD and/or tics. In some cases, a child can experience constant anxiety and fear as well as OCD, whereas other symptoms can gradually disappear. Accurate recognition and diagnosis of PANDAS need to fulfill all of the 5 aforementioned criteria.

## 5. PANDAS vs. PANS

A Pediatric Acute-Onset Neuropsychiatric Syndrome (PANS), is a condition characterized by sudden onset of OCD, tics or disturbed food intake along with various psychiatric distortions, which is similar to PANDAS [71]. Unlike PANDAS, PANS syndrome does not always require previous GAS infection (Figure 1).

Actual criteria of PANS classification include:Rapid onset or recurrence of OCD or abnormalities associated with food intake.Rapid onset of coexisting neuropsychiatric disorders (at least two of them): Increased anxiety levels and/or fear associated with the separation from parentsIncreased motility or motor dysfunctions (including tics and dysgraphia)Behavioral regressRapid decline of grades among children in schoolEmotional lability (irritability, aggression and/or oppositional behaviours)Dysuric symptomsSomatic signs (including insomnia)Present symptoms were not explained and associated with the known neurological or medical conditions [98].

PANS symptoms are induced by the immunological reactions associated with one to several bacteria, or viral infections [84]. These mainly include angina, atypical pneumonia caused by *Mycoplasma pneumoniae*, influenza, upper airway inflammation, or sinusitis [99]. Psychosocial stress may also contribute to the progression of symptoms [54,100].

### The Pathophysiology of GAS Infection and PANDAS

Infectious agents such as GAS could play a role in the immune etiopathogenesis via the activation of various cytokine cascades [101]. Preliminary studies also showed altered IgM, IgA, and IgG subclasses profiles, and the decreased T cell count in PANDAS and Tourette’s syndrome [102]. It has also been reported by Morer et al. that genetic factors may be related to the increase in the D8/17 positive B lymphocytes subpopulation that could trigger the vulnerability to PANDAS [103]. Thus, authors have also suggested that D8/17 could be a potential marker of susceptibility to PANDAS. Sokol et al. observed that PANDAS anorexia nervosa patients were far more likely to have peripheral B lymphocytes expressing D8/17 comparing to patients without PANDAS-characteristics [104].

M protein is a major virulence factor of GAS, which contains brain-cross-reactive epitopes and involved in the pathogenesis of Sydenham’s chorea [105]. M5 and M6 antigens of two separate streptococcal strains were detected in the human brain via immunofluorescence staining patterns [106]. Additionally, M5 was mostly associated with the caudate region, whereas the presence of M6 was more diffuse. This finding suggests that M proteins may trigger the anti-brain antibodies enhancing the severity of symptoms and clinical course [107]. It was observed that neonatal corticosterone contributes to the increase of anti-inflammatory IL-9 in both peripheral and central nervous system in mice model, which may mitigate symptoms of streptococcal infection [108].

Recent research has shown that dopamine levels may contribute to the immunological mechanisms involved in PANDAS pathogenesis [109,110,111]. Altered levels of dopamine may also be an explanation of the decrease of Treg levels and elevations in proinflammatory cytokines [112,113,114,115]. Moreover, it was observed that among PANDAS patients, vitamin D deficiency was significantly more frequent in comparison to the control groups [78]. This finding may lead to the conclusion that vitamin D can be partially associated with the diagnostic criteria of PANDAS or OCD patients, nevertheless, more research associated with this topic should be completed.

## 6. PANDAS and Alterations in the Nervous System

The majority of OCD and PANDAS patients present neuropsychiatric symptoms associated with the impaired functions of either the central or peripheral nervous system, or both [116]. The pathophysiology of PANDAS is mainly associated with the cross-reactions of the antibodies with both streptococcal N-acetyl-beta-D-glucosamine and neuronal lysoganglioside and tubulin [117,118]. PANDAS involves the impairments within the cortico-basal ganglia circuitry [119].

### 6.1. Striatum and Striatal Interneurons

PANDAS patients exhibited significantly enlarged volumes of striatum in comparison to control groups, according to Giedd et al. [62]. The binding of serum antibodies obtained from children diagnosed with PANDAS to cholinergic interneurons in the striatum of mice was significantly higher in comparison to control groups [64]. This finding may present one of the potential loci of the pathophysiology of PANDAS. Cholinergic interneurons are involved in the pathophysiology of tic disorders as well as Tourette’s syndrome, which is indicated by the decreased number of ChAT-positive interneurons in the putamen and caudate [120]. Since the comorbidity of PANDAS and tic disorder is estimated at approximately 60%, the aforementioned finding may be applicable in PANDAS, as well. The reviewed literature also suggests the role of striatal cholinergic interneurons in the progression of PANDAS. The dysregulation within striatal interneurons, including antibody deposition, can lead to impairments in behavior because of the strict association (either direct or indirect) of the interneurons with the basal ganglia [121]. Moreover, it was reported that cholinergic interneurons express D2 dopamine receptors, which are also involved in the binding of antibodies present in PANDAS [122,123,124].

### 6.2. Microglia

Recent studies showed the relevance of microglia in the etiology of PANDAS. Microglia cells play a major role to mediate immune responses in the central nervous system. Nevertheless, it was shown that besides immunological reactions, microglia are involved in brain development, homeostasis, brain plasticity, and adult neurogenesis [125,126]. PET imaging with the usage of 11C-[R]-PK-11195, which is a ligand that binds to the transporter protein expressed by active microglia, showed the increased striatal volumes of children with PANDAS [65]. Furthermore, 11C-[R]-PK-11195 binding was far more intensified in the striatum of children with PANDAS compared to the group of healthy children. Interestingly, the inflammation was more broadly spread and intense mainly in the bilateral caudate and lentiform nucleus. Repeated intranasal GAS infections could result in an elevated number of fCD68+/Iba1+ activated microglia in the glomerular layer of the olfactory bulb [127]. Moreover, it was also hypothesized that GAS antigens could be presented to Th17 cells by the means of the microglia. Synaptic pruning may be elevated in PANDAS, nevertheless, recent research is still restricted to animal models [65].

### 6.3. Basal Ganglia and Antibodies

The first evidence of the presence of antineuronal antibodies in humans was reported by Husby et al. (1977), in the case of Sydenham’s chorea [128]. The presence of antineuronal antibodies and their impact on Sydenham’s chorea was supported by Kirvan et al. (2003) [129,130,131]. It was shown that monoclonal antibodies signal neuronal cells and stimulate them to further release of dopamine and activate CaM kinase II [129]. Several animal models demonstrated behavioral symptoms induced by the deposition of antibodies in the brain and basal ganglia in particular after the immunization with GAS or M18 strain of *S. pyogenes* [132,133]. Furthermore, it was proven that recipient mice or rats may develop PANDAS or Sydenham’s chorea by means of the anti-streptococcal antibodies transfer [134]. Animal models showed that antibodies were deposited in the basal ganglia and thalamus specifically, but not in the cerebellum or hippocampus. In a study of a human Sydenham’s chorea, Cox et al. (2013), demonstrated that putting human V genes for the chorea antibody into transgenic mice, which were then expressed in their B cells, produced the antibodies in serum of the mice [123]. The antibodies penetrated the blood-brain barrier and targeted the basal ganglia, substantia nigra or the ventral tegmental area.

The presence of anti-basal ganglia antibodies in sera of PANDAS patients is a highly specific and sensitive marker in comparison to control, non-affected groups [135]. Antineuronal antibodies that may contribute to the progression of PANDAS symptoms include antipyruvate kinase antibodies, anti-dopamine receptor antibody, and antilysoganglioside GM1 antibody [70,117,129,136,137,138]. In a study performed by Pavone et al. (2004), anti-basal ganglia antibodies were present in nearly two-thirds of patients affected by PANDAS [139]. Additionally, the samples of sera that were obtained from PANDAS patients showed significantly higher activation of CaM kinase II in comparison to healthy controls [140]. The higher activation of CaM kinase II can be associated with the progression of choreic movements in affected individuals. Interestingly, those children who have developed a sore throat due to the streptococcal infection and were treated with antibiotics such as penicillin or azithromycin, showed that they were not at the higher risk of the development of tics or obsessive-compulsive disorders [51].

In general, OCD children present with enlarged volumes of the corpus striatum and basal ganglia, as well as an increased number of antineuronal antibody titers [63]. The size of basal ganglia is strictly associated with the total number of antibodies present during GAS infection [141,142]. Furthermore, Cabrera et al. found a greater volume of grey matter within basal ganglia and a reduced volume of white matter in this area [143]. It was reported that the infection of GAS is correlated with the production of antineuronal antibodies. The presence of streptococcal antineuronal antibodies can be associated with the neuronal damage as these antibodies interact with structures of the nervous system, as a result of the antibody mimicry [12,144,145,146,147]. There is no significant correlation between the size of the basal ganglia and the severity of symptoms among the studied groups. The neuroimaging studies are consistent with the pathological immune reactions of the antibodies reacting with the basal ganglia leading to their altered volumes. Even some observable variations of the basal ganglia volumes are present and observable during examinations, the knowledge is still restricted to the potential functional changes, which may be present within basal ganglia due to the aforementioned impairments. Therefore, there was no study strictly associated with the alterations of the basal ganglia functions specifically; it could be the potential area of interest regarding future studies associated with PANDAS.

### 6.4. Other Structures in CNS

Giedd et al. (2000), demonstrated that children with streptococcal OCD present with increased sizes of the caudate, putamen, and globus pallidus, whereas the average size of thalamus and total cerebrum remains unchanged [62]. Further studies proved the aforementioned results showing that all of the previous structures besides thalamus are enlarged in the case of PANDAS patients. The caudate and putamen are specifically enlarged most extensively in the acute phase of PANDAS. This research may also indicate the potential localization of the pathophysiological mechanisms of this syndrome. Besides the alterations that were observable using Magnetic Resonance Imaging (MRI), the study with the usage of Positron Emission Tomography (PET) showed that PANDAS patients tend to present with a higher percentage of binding potential values in bilateral caudate and lentiform nuclei compared to the control group [148].

## 7. The Characterization of Gut Microbiota

### 7.1. The Impact of Gut Microbiota on the CNS

There is an increasing amount of research that presents the relationship between the gut microbiome and the development of psychiatric disorders [149,150]. The most recent data indicates that gastrointestinal homeostasis significantly affects the development and functioning of the central nervous system, which presents the so-called microbiota–gut–brain axis as the new paradigm in neuroscience [151]. Neuropsychiatric disorders where the etiopathology can be related to the gastrointestinal microbiota include anxiety, autism, anorexia nervosa, Parkinson’s disease, Alzheimer’s disease, ADHD, schizophrenia, alcohol dependence, bipolar disorders, or migraine pain [82,100,152,153,154,155,156,157,158,159,160,161]. Interestingly, the microbiota-gut-brain axis starts to develop during the intrauterine period of fetal development. The axis is affected by both intrauterine and extrauterine factors including antimicrobial treatments of the mother, vaccinations, increased contact with chemicals, diet, type of child delivery, feeding habits during infancy period, as well as exposure to bacterial, viral or parasite infections and maternal stress during pregnancy [162,163]. Moreover, the stages of gut microbiome development occur simultaneously with the neurodevelopment. The connection between gut microbiota and central nervous systems occurs by several pathways, which involve microbial metabolites (short-chain fatty acids), immune cells, tryptophan metabolism along with neural (vagus nerve), and endocrine pathways (the hypothalamic-pituitary-adrenal axis) [164,165]. Bacterial commensals in the gastrointestinal tract are also responsible for the production of γ-aminobutyric acid, which is the main inhibitory neurotransmitter in the central nervous system and may induce neuropsychiatric symptoms [166,167,168,169,170,171,172]. Dopamine, noradrenaline, and histamine, as well as brain-derived neurotrophic factor (BDNF) levels, can also be affected by gut microbiota [173].

### 7.2. Alterations in Gut Microbiota in PANDAS

Quagliariello et al. performed a study to investigate the bidirectional connection between the gut microbiota and the central nervous system in PANDAS patients [66]. A group of 30 PANS/PANDAS patients aged 4–16 (20 males and 10 females) were studied in terms of microbiota that colonizing their gastrointestinal tracts. The group of 4–8 years old patients presented the altered abundance of phylum levels including increased Bacteroidetes and lower Firmicutes. This group of patients also presented with lower distribution of Actinobacteria and the total absence of the TM7 phylum (Saccharibacteria). Regarding family levels, the group of 4–8 years old patients presented with higher levels of Bacteroidaceae, Rikenellaceae, and Odoribacteriaceae. Contrarily, some of the Firmicutes families including Turicibacteraceae, Tissierellaceae, Gemellaceae, and Carnobacteriaceae (Bacilli class), Corynebacteriaceae and Lachnospiracea (Clostridia class) were absent. Nevertheless, within Firmicutes phylum, Turicibacteriaceae, Erysipelotrichaceae, and Lachnospiraceae are considered to be potential characteristic gut microbiota in PANDAS patients even if their distribution was at a minor level. Furthermore, Bacteroidaceae were present greatly in the gastrointestinal tract of PANDAS patients. Moreover, a strong negative correlation between increased levels of Bacteroidaceae and other bacterial families has been stated. On the other hand, PANDAS patients above 9 years old presented with a greater abundance of Peptostreptococcaceae and Erysipelotrichaceae and lowered levels of Rikenellaceae and Barnesiellacea. The discrepancy between gut microbiota in two aforementioned groups may occur because of a different age of PANDAS patients as microbial diversity in the gastrointestinal tract significantly differs throughout the lifetime [174,175,176,177]. Many treatment therapies especially antibiotic drugs significantly affect gut microbiota and function, which constitute a limitation preventing the detection of possible microbial markers associated with PANDAS [178].

According to the authors’ knowledge, it is the first detailed study of gut microbiota specifically in PANDAS patients, therefore more research should be accomplished to compare the results of studies and provide further information.

Nevertheless, as it was stated in this study, alterations between levels of gut microbiota were clearly observable in cases of PANDAS patients of different age. This observation may lead to the conclusion that alterations in the gut microbiota constitute one of the factors enhancing PANDAS development, as well as the potential marker of further diagnostic procedures of this syndrome.

## 8. Genetic Approach to OCD, Including PANDAS

With regards to the genetic approach to the understanding of the pathophysiology of PANDAS, the studies are still scarce and usually limited. Murphy et al. in 2010 reported among examined biological mothers of 107 children with either OCD, tics or both, the rate of autoimmune diseases was 17.8%, whereas in the general population the rate was equal to 5% [179]. This finding may present the potential genetic aspect considering the development and possibly the progression of PANDAS patients, as well since PANDAS is considered to be the subtype of OCD. In other disorders like Tourette syndrome, which also present various intensity and severity of tics, several mutations including Slit and Trk-like family member 1 (*SLITRK1*) gene are present in rare cases [180]. Interestingly, the expression of this gene is strictly associated with the cholinergic interneurons in the adult striatum. Additionally, regarding Tourette syndrome, acetylcholine-related genes seem to correlate with the severity of this disorder [181]. Even though the aforementioned results were not studied in the case of PANDAS patients, it should be considered that the comorbidity of Tourette syndrome and OCD along with OCD and PANDAS is equal to 50% and 60% correspondingly. Though patients with Tourette syndrome present with elevated levels of immune gene expression restricted to basal ganglia specifically, these results are still not clear in the case of PANDAS patients. Mannose-binding lectin (MBL) plays an important role in immune responses. It was stated that GAS show a high susceptibility to bind to *MBL* [182]. Any variant in exon 1 of the *MBL2* gene increases the probability of PANDAS [67]. Interestingly, recurrent infections may be associated with the polymorphism of codon 54 of the *MBL2* gene [183]. *MBL2* polymorphism is also suspected to play a role in the pathogenesis of rheumatic heart disease and rheumatic fever [184]. Therefore, the *MBL2* gene is suspected to be one of the diagnostic markers of PANDAS. According to Luleyap et al., TNF-α −308 AA polymorphism can be regarded as a molecular indicator of PANDAS [68]. Altered levels of TNF-α can be associated with exacerbation of PANDAS symptoms and facilitating autoimmune responses within the central nervous system. There is not much research done regarding genetic aspects associated with PANDAS. Even though, several pieces of research indicate the relationships between particular genetic variations and the onset of PANDAS.

## 9. Treatment of PANDAS

### 9.1. Antibiotic Therapy

Treatment of PANDAS is to a large extent restricted to its severity including symptoms, exacerbations, and trajectory [79]. PANDAS treatment therapy mainly includes psychoactive drugs, immunotherapeutics with steroids, antibiotics, plasmapheresis as well as intravenous immunoglobins [80,185,186,187,188]. With regard to the proper treatment of PANDAS, it should be considered that the application of penicillin or other antibiotics alone does not constitute the only way of treatment of this disease [171]. It is because penicillin is useful only in the cases of active streptococcal infection to eradicate bacteria. Nevertheless, in cases of persistent and active infection of GAS, immediate elimination of the infection source should be provided to minimize the severity of PANDAS. Moreover, the most recent research showed that neuropsychiatric exacerbations can be minimized by the application of penicillin V and azithromycin even after the one-year duration of PANDAS [72]. The main antibiotic used in the streptococcal eradication is penicillin, but additional drugs such as amoxicillin or azithromycin are also applied [79]. Some research suggests that azithromycin can also be helpful in the alleviation of PANDAS symptoms, nevertheless, its application should be strictly controlled because of the potential side effects mainly in the form of cardiac risks [73]. According to the most recent research, it is more common to provide treatment in the form of antibiotics rather than the application of psychiatric treatments including cognitive-behavioral therapy [189,190].

### 9.2. Psychoactive Drugs

Treatment of PANDAS is a complex process that includes not only psychoactive medications but additionally behavioral-cognitive therapy which is offered mainly to those children who present with high levels of distress and anxiety [186,191]. The cognitive-behavioral intervention seems to be a less invasive and expensive way of treatment in comparison to antibiotic therapy or selective serotonin reuptake inhibitors (SSRIs) [77]. Even though, clinicians should take into consideration the possibility of combining the aforementioned techniques to obtain better clinical outcomes among PANDAS patients. Antipsychotic drugs including risperidone can be applied in cases of severe behavioral symptoms [74]. Unfortunately, only a small number of studies researched the effects of psychiatric therapies in PANDAS. It was stated that among examined patients, the usage of SSRIs resulted in the improvement of the OCD symptoms [192,193,194]. Patients with PANDAS who also present with the ADHD symptoms can be prescribed atomoxetine [195]. One of the relatives of benzodiazepine, lorazepam, shows the potential in PANDAS treatment since it improves the motor activity, expressive language, as well as the posturing of patients [75]. Such symptoms like mood lability or compulsion to void, which are not alleviated by lorazepam, appeared to be improved by the means of plasmapheresis application.

### 9.3. Nonsteroidal Anti-Inflammatory Drugs

In the observational study performed by Spartz et al. in 2017, it was reported that the usage of the nonsteroidal anti-inflammatory drugs (NSAIDs) in PANDAS pediatric patients, provided different and in some cases contradictory results. Approximately 31% of examined patients show improvements regarding OCD and tic symptoms. Contradictorily, 35% of patients present the aggravation of tics severity and patients’ mood. So far, it is unknown why the examined patients present different therapeutic effects of NSAIDs. However, both prophylactic and early usage of NSAIDs is associated with the shorter duration of the exacerbation of PANDAS symptoms [196]. Because of the wide range of the results which are available at the actual state of knowledge, nevertheless, additional research trials should be done to specify the role of NSAIDs application in PANDAS patients.

### 9.4. Tonsillectomy

There is only a small number of reports regarding the application of tonsillectomy to eradicate symptoms of PANDAS. In 2011, Alexander et al. reported a case of a 9-year-old boy with PANDAS who presented with frequent and recurrent streptococcal infections, which were resolved just after tonsillectomy [197]. The positive clinical outcomes after tonsillectomy could be observable not only because of the removal of tonsils but also because of proper medications provided just after the tonsillectomy procedure [198]. Tonsillectomy is a reasonable procedure, mainly in those cases when the usage of antibiotics does not bring positive clinical outcomes and in the recurrent GAS-associated tonsillitis. This can also be an alternative way of treatment in the severe cases of patients who present with multiple resistance to several groups of antibiotics. In the study performed by Demesh et al. in 2015, 33% of children who had a tonsillectomy, presented with the complete resolution of neuropsychiatric symptoms [199]. In those children, whose symptoms were not resolved completely, it was observed that the severity of symptoms was significantly reduced. Even though, there are still cases of increased morbidity even after tonsillectomy procedure [198]. Due to the limited research in this area, it is still unknown whether tonsillectomy should be treated as a preventive procedure of PANDAS symptoms, or rather than the second-line therapy for those patients who present with a poor clinical outcome during antibiotic treatment.

### 9.5. Other Treatment Strategies

There is still, there is a lack of consistent evidence in PANDAS treatment with antimicrobials and immunotherapy as well [200]. Intravenous immunoglobulin therapy (IVIG) was observed to decrease the number of pathogenic antibodies in PANDAS patients alleviating OCD symptoms [201]. It is advised to follow the immunotherapy as well, which could decrease the side effects of neuroinflammation and autoimmunity associated with streptococcal infection. Milder symptoms of the disease can be treated with the usage of non-steroidal anti-inflammatory drugs, whereas more severe symptoms are treated with intravenous immunoglobulins. Plasma exchange can be applied in cases when symptoms are of such intense severity that may be life-threatening [202]. Application of plasmapheresis may also lead to the stabilization of the basal ganglia volumes, which are altered during the disease [141,203,204,205]. Since vitamin D metabolism has been observed to play a role in the pathogenesis of PANDAS, supplements of this vitamin can be useful in the reduction of psychiatric symptoms of patients [206,207].

## 10. Conclusions

Even though the concept of PANDAS was first introduced in 1998 by Swedo et al. it is still difficult to unequivocally distinguish its symptoms from a typical OCD or a tic disorder. To provide the most accurate diagnosis, several aspects of PANDAS etiopathology should be taken into consideration, including the infection of GAS, rapid onset, genetic predispositions, environmental factors, or even the coexistence of other diseases—mainly psychiatric ones—which can enhance PANDAS symptoms [208]. Moreover, it should be noted that immune factors are not only the markers of the disease, but they seem to be pathogenic, as well [140]. PANDAS has a male predominance according to the studies. With regard to alterations in the nervous system in PANDAS patients, several important observations have been reported. Various neuronal proteins, including tubulin, lysoganglioside, and dopamine receptors are affected by the antibodies produced in the course of PANDAS. As a consequence, some alterations of cortico-basal ganglia circuity are observed [209]. The deposits of antibodies are also accumulated in the striatal interneurons. PANDAS patients present significantly enlarged volumes of corpus striatum, caudate, putamen, globus pallidus, and basal ganglia. Since the activated microglia were found in close proximity to CD4+ T cells, it was assumed that local microglia might present GAS antigens to Th17 cells [65]. Besides manifestations from the central nervous system, noticeable changes in the presence and distribution of specific microbiota in the gastrointestinal tract were observed in PANDAS patients [66]. This might have an indirect impact on behavior due to the reduced production of various metabolites involved in brain functions. Further, exon variants of the *MBL* gene and altered levels of TNF-α increase the susceptibility of PANDAS [67,68]. The treatment of PANDAS syndrome is still not obvious and specified, however, there are several alternatives that show promising results. Among them are antibiotics, immunomodulatory drugs, psychiatric drugs, and behavioral interventions.

## Figures and Tables

**Figure 1 ijms-21-01476-f001:**
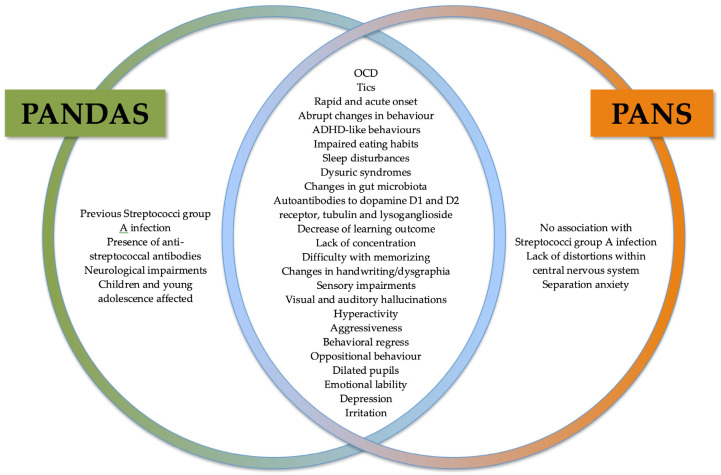
Common and distinct symptoms characteristic to PANDAS and PANS.

**Table 1 ijms-21-01476-t001:** Characteristics of Pediatric Autoimmune Neuropsychiatric Disorders Associated with Streptococcal Infections (PANDAS), Pediatric Acute-Onset Neuropsychiatric Syndrome (PANS), Sydenham’s chorea, and Tourette syndrome.

Psychiatric Syndrome	Symptoms	Changes in CNS	Changes in Microbiota	Genetic Alterations/Mutations	Presence of Antibodies	Available Treatment
**PANDAS**	OCD, tics, obsessions, compulsions [60], anxiety, agitation, aggression, insomnia, impulsiveness, emotional lability, depression, hyperactivity, suicidality [54], dysuric symptoms, dilated pupils, inattention, behavioral regress, dysgraphia, visual and auditory hallucinations, irritation [61]	Enlarged striatum, caudate, putamen, basal ganglia and globus pallidus [62,63], dysregulation within striatal interneurons [64], overactivation of microglia [65], increased number of antineuronal antibody titers [63]	Higher percentage of Bacteroidetes, Rikenellaceae and Odoribacteriaceae; lower of Firmicutes and Actinobacteria; absence of Saccharibacteria and Turicibacteraceae, Tissierellaceae, Gemellaceae, and Carnobacteriaceae (Bacilli class), Corynebacteriaceae and Lachnospiracea (Clostridia class) [66]	*MBL2* [67], TNF-α −308 AA polymorphism [68]	Autoantibodies against: dopamine D1, D2 receptor, tubulin, lysoganglioside, antipyruvate kinase [69,70], calcium calmodulin dependent kinase II [71]	Penicillin V, amoxicillin, azithromycin [72,73], steroids, antipsychoactive drugs (risperidone) [74], immunotherapeutic, plasmapheresis [75], intravenous immunoglobulins [76], cognitive-behavioral therapy, SSRIs [77], vitamin D supplementation [78]
**PANS**	OCD, tics, impaired food intake, anxiety, separation fear, motor dysfunctions, dysgraphia, behavioral regress, regress in school outcome, emotional lability, irritability, aggression, oppositional behaviours, dysuric symptoms, insomnia	ND	Higher percentage of Bacteroidaceae, Rikenellaceae, and Odoribacteriaceae lower level of Firmicutes, absence of Turicibacteraceae, Tissierellaceae, Gemellaceae, and Carnobacteriaceae (Bacilli class); Corynebacteriaceae and Lachnospiraceae (Clostridia class); Bifidobacteriaceae (Actinobacteria) and Erysipelotrichaceae [66]	ND	Autoantibodies against: dopamine D1, D2 receptor, tubulin, lysoganglioside, calcium calmodulin dependent kinase II [71]	First-line antibiotics: penicillin, amoxicillin Alternative antibiotics: azithromycin, cefadroxil, cephalexin, cefpodoxime Other: Intravenous immunoglobulin, therapeutic plasma exchange, corticosteroids, SSRIs, non-SSRI-antidepressants, ADHD medication, antipsychotics, anxiolytics, mood-stabilizers, cognitive-behaviour therapy [79,80]
**Sydenham’s chorea**	OCD, tics, ADHD, generalized anxiety, mood disorders, psychotic features, emotional lability, irritability, regressive behaviour, separation anxiety, panic disorder, phobias, social phobia, agoraphobia, executive dysfunction [81]	Permanent basal ganglia damage, dysfunction in the connection between the basal ganglia and the superior colliculi [56]	ND	ND	Autoantibodies against: dopamine receptor D2, anti-basal ganglia antibodies [82]	Penicillin, neuroleptics (risperidone, haloperidol), sodium valproate, corticosteroids, plasma exchange, intravenous immunoglobulins, dopamine depleters (tetrabenazine, deutetrabenazine, valbenazine), dopamine antagonists [83]
**Tourette’s syndrome**	OCD, tics, ADHD, depression, bipolar disorder, anxiety, personality disorder, learning disability, speech and language disorder, intellectual disability, trichotillomania, sleep disorders, pathologic nail-biting, simple phobia, social phobia, agoraphobia, impulse control disorders (intermittent), explosive disorder, self-injurious behavior, impulsive-compulsive sexual behavior [84,85]	Increased functional connectivity between the basal ganglia nuclei (right and left caudate, putamen and left pallidum) and cortex (superior temporal gyrus, medial temporal gyrus, paracingulate gyrus, precuneus, angular gyrus, insular cortex) Increased connectivity between left thalamus and the cortex (right planum temporale, right superior temporal gyrus) [86,87]	ND	*CELSR3*, *FLT3*, *COL27A1*, *SLITRK1*, *SLITRK5*, *HDC*, *NRXN1*, *CNTN6*, *DRD3*, *GDNF*, *KCNJ5*, *AADAC* [88,89]	Antibodies against: rheumatogenic GAS serotypes (M12, M19), anti-neuronal antibodies, antibodies to membrane isoforms of glycolytic enzymes (aldolase C, enolase, pyruvate kinase M1), anti-HCN4 antibodies [90,91,92]	First line: aripiprazole, sulpiride, risperidone Second line: clonidine, guanfacine Third line: topiramate, pimozide, tetrabenazine Fourth line: clonazepam, haloperidol Also: botulinum toxin, SSRIs (sertraline), tricyclics (clomipramine), stimulants (methylphenidate, dextroamphetamin), non-stimulants (atomoxetine, clonidine, guanfacine) Other: behavioural treatments, deep brain stimulation [87]

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
