# Peer review of "Alterations in the Nervous System and Gut Microbiota after β-Hemolytic Streptococcus Group A Infection—Characteristics and Diagnostic Criteria of PANDAS Recognition"

_ijms, 2020, doi:10.3390/ijms21041476_

Round 1

Reviewer 1 Report

I thank the authors for their attention to my previous comments while revising the manuscript.  There are still a few minor grammatical changes that can be made but the manuscript is greatly improved in its current form.  One more thorough reading for grammar should yield an acceptable manuscript.

Author Response

Jacek Baj

Medical University of Lublin

Department of Human Anatomy

Jaczewskiego 4

20-090 Lublin, Poland

[email protected]

16th February 2020

Dear Reviewer,

Thank you very much for reviewing our manuscript, providing extremely important comments which have greatly improved this paper. We are very grateful for the commitment you have provided while reviewing this manuscript.

We are pleased to submit the manuscript entitled “Alterations in the nervous system and gut microbiota after β-hemolytic Streptococcus group A infection - characteristics and diagnostic criteria of PANDAS recognition’’.

According to your comments, the manuscript was checked once again by one of our native speaker colleagues and several improvements of English grammar were made throughout the text.   

We would like to thank you once again for your effort, time and extremely precious comments.

We wish you all the best!

Sincerely,

Jacek Baj

on behalf of the authors

Reviewer 2 Report

The manuscript is now fluent and appears to be worth reading

There are still few typos. Here are some:

Line 23                                 X articles >>>>  please specify the number

Line 792                               Interestingly is misspelled

Line 1231                            life-threatening is mispelled

Author Response

Jacek Baj

Medical University of Lublin

Department of Human Anatomy

Jaczewskiego 4

20-090 Lublin, Poland

[email protected]

16th February 2020

Dear Reviewer,

Thank you very much for reviewing our manuscript once again. We are very grateful for the commitment you have provided during the revision of this manuscript and appreciate your precious comments.

We are pleased to submit the manuscript entitled “Alterations in the nervous system and gut microbiota after β-hemolytic Streptococcus group A infection - characteristics and diagnostic criteria of PANDAS recognition”.

The manuscript has been checked once again in terms of English by one of our native speaker friends and we hope that now it is much more improved.

Additionally, we have removed the typos:

Line 23 – we have removed the ‘X’ just leaving the “Articles” word alone

Line 792 – we have changed the typo into ‘Interestingly’

Line 1231 – we have changed the typo into ‘life-threatening’

We are sorry for those misspellings and hope that now the manuscript is improved.

We wish you all the best!

Sincerely,

Jacek Baj

on behalf of the authors

This manuscript is a resubmission of an earlier submission. The following is a list of the peer review reports and author responses from that submission.

Round 1

Reviewer 1 Report

Comments for the authors of the International Journal of Molecular Sciences number ijms-650339:

The authors of the International Journal of Molecular Sciences manuscript “Alterations in the nervous system and gut microbiots after b hemolytic Streptococcus group A infection- characteristics and diagnostic criteria of PANDAS recognition”, provide thorough review of the current literature associated with the neurological disease syndrome known as PANDAS.  This review summarizes what is known about the pathogenesis of the disease, starting with pharyngitis, presentation of the symptoms and diagnosis criteria, comparison with other infection-induced neurological syndromes, known effects of PANDAS on cells of the nervous system, contributions of the microbiota, genetic associations, and treatment.   While the information presented in the manuscript is valuable, and the literature search is clearly extensive, there are a number of major and minor comments that the authors should consider during revision.  These comments are listed below. 

General Comments:

The manuscript should be edited, with attention paid to sentence and paragraph structure. The text is presented in a very dense manner, with paragraphs that extend as long as 78 lines.  This limits the reader’s ability to focus on the key points of each paragraph.  The suggestion is to break the text into paragraphs that focus on 1-2 key points and to re-design the text around this outline.  The use of sub-headings may help in this effort. Inclusion of additional tables and/or figures may help the authors make specific points more clearly for the reader. A figure like Figure 1 is very helpful. Table 1 would benefit from a revision. Maybe the key characteristics of all diseases could be presented as the table headings, and the syndromes with those characteristics could be indicated within the cells of the table. The symptoms that are unique to each syndrome could then be indicated using either a footnote or in the text itself.

Specific comments

The text on line 64 that begins with “The symptoms of rheumatic fever appear…” could be presented earlier within that paragraph. The choice to use the abbreviation GAS on line 82 seems a bit late in the manuscript, and it could probably be introduced on line 40. The word “ani-basal” on line 115 seems like a mis-spelling. In Table 1, what does “ND” stand for? The sentences that begin on line 287 and ending on line 290 should be revisited to make sure they are presented in the correct order. On line 315, the word “antiantibodies” seems like a mis-spelling. The word “fatal” on lone 331 seems to be incorrect.

Reviewer 2 Report

The authors, Baj, et al, are to be commended for putting a comprehensive review together on PANDAS entitled “ Alterations in the nervous system and gut microbiota after b hemolytic Streptococcus group A infection – Characteristics and diagnostic criteria of PANDAS recognition”. The review discusses rheumatic fever and PANDAS including the history of disease and diagnostic guidelines and common and distinct symptoms characteristic of PANDAS and PANS. They have also compiled some excellent references and information but some of the references are either misused or incorrect and need adjusting. The authors focus on the clinical features of disease, the alterations in the nervous system, gut microbiota and genetic aspects of these diseases. The authors must improve the review by addressing the following comments.

The history of rheumatic fever is important in any review related to group A streptococcal sequelae. In the review they have references to older work in rheumatic fever but did not seem to reference the more current literature. For example, the review in Nature Reviews Disease Primers from 2016 Carapetis J, Beaton A, Cunningham MW , Guilherme L, et al is the most comprehensive and most current overall high impact review that they should definitely reference. This review in itself has many of the most important references which seem to be lacking in the first section of the PANDAS review where the authors are discussing acute rheumatic fever. Other reviews that should be cited are Guilherme, L in Lancet in the past year or so and also several Cunningham, MW reviews in Current Topics in Rheumatology in 2012 and International Reviews in Immunology approx. 2014. Further, the American Heart Assn guidelines for diagnosis and treatment of rheumatic fever probably should also be referenced. This is found in the American Heart Assn Scientific Statement on rheumatic fever in the reference Circulation. 2015;131:1806-1818. DOI: 10.1161/CIR.0000000000000205 In the section on PANDAS, the first sentence is about the first 50 cases discovered and reported by Susan Swedo and it needs to be emphasized that many of these cases if not all had piano playing choreiform movements of the fingers and toes. These signs of PANDAS may be difficult to notice. Even Sydenham chorea can go unnoticed by parents and doctors who are not familiar with the symptoms and signs of these diseases and this is worth pointing out. The discussion about the antibodies against the group A streptococcus as well as its M protein are not very clearly written and will lead to confusion. These statements are at the top of page 3 in regard to M5 and M6 proteins. Were animals immunized with the strains M5 or M6 or were they the infecting strains in the humans with the disease? The statement is not very clear on this. Lines 1-4 from top of page 3. One of the most cited animal studies of the immunization with the M proteins leading to reactivity with the brain is by Bronze et al. This reference would be good to include at this point in the review on page 3 top of page. The first evidence of antineuronal autoantibodies in humans was found by Husby, Zabriskie et al in an important paper about the antibodies deposited on neuronal cells in the human brain. A more current study of these anti-brain antibodies in Sydenahm chorea was in 2003 when a human mAb was reported from Sydenham chorea which signaled neuronal cells to produce elevated levels of dopamine and activate CaMKII in a neuronal cell line(Kirvan et al Nature Medicine 2003) This reference needs to also be added to the review. None of the mouse or rat models of PANDAS or SC were discussed or referenced in the article. This is an oversight by the authors. These references include study of immunization of the mouse with group A streptococci(Hoffman, M Hornig et al) and immunization of the Lewis rat with an M18 strain of S pyogenes(Brimberg L, D Joel, MW Cunningham et al . Further studies in both of these animal models led to the proof that the anti-streptococcal antibodies would transfer the disease of PANDAS or SC to the recipient mice or rats (Yaddanpudi, Hornig et al, and Lotan, D, Joel D, Cunningham MW et al). All of these animal models were important in demonstrating behavioral symptoms associated with antibody deposition in the brain and basal ganglia; and that they could also be seen when the antibodies were passively transferred. The antibodies deposited in the brain basal ganglia, thalamus but not the cerebellum or hippocampus. A study of a human Sydenham chorea derived monoclonal Ab demonstrated that when the V genes for the chorea antibody were put into Tg mice expressing these chorea Ab V gene in their B cells produced the antibody in serum of the mice. The human chorea antibody penetrated the brain and basal ganglia after breaking the BBB in the Tg mice. In the Tg mice, the Ab targeted the Basal ganglia and specifically the substantia nigra or the ventral tegmenal area or VTA. The reference to this Tg mouse study is Cox, CJ, Sharma, Kovoor,A Cunningham, MW; J Immunology 2013) and should be added to the review. Since the authors are discussing both PANDAS and SC, these references are important to the overall review and the addition of more current knowledge. One of the points that the authors do not realize is that the antibodies that lead to rheumatic fever, PANDAS etc are reactive with the dominant epitope of the group A streptococcal carbohydrate N acetyl b D glucosamine as described in references including Galvin et al J Clinical Investigation for the rheumatic fever and carditis and then Nature Medicine article Kirvan et al. These are minor points but there is mention of the streptococcal M protein and it was after those studies that it was realized from the human mAbs that the epitope was GlcNAc of the group A carbohydrate. This point and references should be added in the PANDAS and rheumatic fever sections. On page 3, lines 14-16 states “PANDAS is mostly referred to as a group of either children or adolescents and its prevalence is significantly higher in males than in females. The reference given is a case report of a single male boy and is ok to leave in the review but the authors must use references of larger numbers rather than a report of ONE case. A reference to studies by Sue Swedo or Tanya Murphy on this point would be appropriate because they would have used a larger group of children to determine that this statement is true and supported by the data in the reference. One case is not enough for this statement. Midway down on page 3 lines 21-23 from bottom of page, the statement about the D8/17 antibody is not exactly correct. The antibody did not bind to malignant B lymphocytes, but lymphocytes were investigated from cases of rheumatic fever and a larger proportion of the individual’s B cells were stained if they had rheumatic fever or were at risk or potentially susceptible to rheumatic fever. This work began using a multiparous serum from a mother who had antibodies that seemed to recognize B cells in rheumatic fever or those who might be at risk. That article may be more explanatory and the senior author would be John Zabriskie. A monoclonal was prepared after the serum was exhausted but actually never game exactly the same results but may be useful. There is confusion about what it is binding on the B cells and if it is useful in detecting risk. Thus, the biomarker had been investigated in PANDAS. The authors should go back to the PANDAS study and clear up the incorrect information in the text about this antibody. It is not clear that the authors understand this information. The section on the diagnostic guidelines is important but does not have all the references to the complete set of guidelines which should be added to the review.

These references include 5 articles which they reference at least 2 of the set. The first article published is a consensus article written from a meeting at Stanford University and it discussed the diagnosis of PANDAS and PANS and was a consensus published in 2013 in the Journal of Child and Adolescent Psychopharmacology or JCAP. This article should be added to the references. Then in a few years, there were guidelines published in JCAP for the treatment and this fell into an overview and into 3 main categories which all should be main references in the section on guidelines. All 4 referencdes should be in the set of references for the artoicle and they should be used for their treatment guidelines or explain if and why they are deviating from them. These references include one that they do reference Thieneman et al for the psychiatry and behavioral interventions guidelines. However, there were 3 more articles on the treatment guidelines which must be added to the references and they also must be used and the guidelines stated must give these references and also any other references used to generate the guidelines in this review. The Thieneman reference #49 in the current review but the rest of the guidelines were missing from the review and seem to be needed to support the guidelines they are proposing or they need to explain why their recommendations are different from these published guidelines. It is impossible for this reviewer to go point by point in each set of guidelines. The way they are set up in tables seems to make them difficult to go through.  The authors will have to decide the best way to show them and refer readers to the guidelines that are already published and discuss new additions etc which they feel are warranted to add. For the most part the lists of symptoms are correct but the authors need to verify that they are in the guidelines already published and refer the readers to these guidelines.

In addition, the references to the guidelines should be listed directly in the tables and titles etc. such as the reference #49(part 1) should be put into the first sentence of the PANDAS guidelines but more than just the psychiatric reference to Thieneman is needed unless they state that they are not discussing any of the other guidelines except the Psychiatric part of the disease. The authors show a list in the figure 1 for example of the anti-neuronal autoantibodies to dopamine D1 and D2 receptor, tubulin and lysoganglioside. The antibodies as well as the other things listed all need references after them. For example the antibodies references may need several such as the Kirvan et al in nature medicine reference for lysoganglioside and the CaMKII assay or the J Neuroimmunology reference on PANDAS by Kirvan et al, the reference to the tubulin Kirvan et al in the Journal of Immunology, a reference to the D2R in Cox, Sharma, Kovoor and Cunningham and a reference to the entire group including the D1R which is published Cox et al in JCAP 2015. The guidelines must have the references to back up the diagnosis and treatment that they propose in the review. It is a important work to get guidelines correctly referenced in tables or figures and all referenced to articles that support it. There are too many unreferenced lists etc. The lists in Table 1 must be accompanied by a list of all the references to items used in the Tables. No one will be able to figure out where their information came from. On page 9 approx lines 19-20 from the top of page, needs something there added in for the pathophysiology of Sydenham chorea to state “Kirvan et al demonstrated antibodies binding to the caudate putamen in basal ganglia in SC and PANDAS” Nature medicine 2003 and Journal of Neuroimmunology 2006. Line 17 from bottom, Reference 65 does not appear to be the correct reference. This is Pittenger, Williams, Swedo et al about the cholinergic neurons and then D2 reference to Cox, Sharma, Kovoor, Cunningham et al in the Journal of Immunology 2013 . The D2 is from one reference and the cholinergic neurons are from the Pittenger, Williams reference which is recent. 65 is just a reference pulled out of the literature and does not really support that statement. Line 7 from the bottom of page 9 needs the word cells after Th17. Top of page 10, line 2, reference 73 needs to be supplemented with the referfences that actually provide the information about antibody mimicry. These are the Nature Medicine Kiran et al 2003, Carapetis, Nature Reviews diseases primers, 2016, Journal of Neuroimmunology Kirvan et al 2006, Current Opin in Rheumatology Cunningham MW. This reference 73 is not a mimicry reference. On page 10 line 17 or 317 from top of page, the reference 75 does not provide reference to all those antibodies so the group of references should be added inc;luding the Nature Medicine Kirvan et al 2003, anti-dopamine receptor antibodies, anti-lysoganglioside antibodies—all came from specific references that are not provided here. Kirvan et al J Neuroimmunol 2006 and the anti=pyruvate kinase was not clear because the Calcium calmodulin dependent protein Kinase II activation by antibodies or sera is the antibody mediated signaling believed to cause the symptoms in the disease since these are functional antibodies. However, the CaMKII did not seem to be mentioned here or any of these referfences to the list of the antibodies given on page 10 Lines 315-17. Animal model studies mentioned previously with tehreferences are also important in supporting the findings from PET imaging or antibody removal by plasma exchange. The Perlmutter et al study of Plasma exchange and IVIG treatment is a reference that the authors did not seem to know but should be used. Study of the antimal model or passive transfer led to changes in the behavior and wsas associated with Ab ddeposition in the basal ganglia and the Abs were the same(anti-D1R and D@R and CaMKII activation by rat serum) as those reported from human tissues in the Brimberg et al(J Neuropsychopharmacology)and Lotan et al articles(Brain Behavior and Immunity). These sections are written with some of the correct ideas but the authors do not really know where they came from as far as references. In section 6, the first few sentences are not written in good English. Fetal is misspelled in the first section of the microbiota section. There are probably a lot of references which are not listed in the microbiota section as only a few are given for a large amount of data discussed. In the section on genetics, Tourettes is referred to as a malignancy and this term seems to be used a lot. The authors clearly must know what Tourettes is but this is a very serious flaw in the review which will be read by individuals trying to learn about these diseases. Malignancy is cancer, Tourettes is a neuropsychiatric disease!! A reference and justification for stating that the MBL gene predisposes to PANDAS. This is not even referenced in the article and this reference is needed for this to bed a credible statement. Just because it is a potential gene risk in ARF, it is not necessarily in PANDAS if studies have not been done. References to PANDAs must be given. Reference 95 is for helicobacter pylori…not group A streptococci…so whatever the authors are trying to say is not clear at all in this section! The authors need to explain the MBL gene and why it would be important. Any gene should be explained as to why it would be important In the discussion section, the point about the rapid onset of PANDAS needs to be strengthened and maybe this should be stated several times throughout the review. The Kipnis reference should be added in the discussion about the T regulatory cells. The

Do the authors have references where the T reg are decreased in PANDAS patients? This would need to be added to the discussion. Is their a decrease in dopamine in PANDAS? Please give directly these references in the discussion.

Reviewer 3 Report

The topics addressed by this study is timely and interesting and the literature seems to have been exhaustively quoted.

That said, the manuscript has many flaws. First of all, although English is formally correct (at least from a grammatical point of view, apart from a few verbal forms), the wording is in many instances inappropriate (e.g. the word malignancy is not appropriate in this context), the sentences are not hierarchically organized and some phrases sound altogether weird. Whether it is for the fact that the Authors’ mother language is not English or for other reasons, also the matter appears to be not properly distributed, thus leading to difficulty in drawing conclusions. In particular, the connections between autoimmune responses, which should be the main focus of the review, neuro-inflammation and swelling of certain brain areas is not brought to the attention of the reader. Moreover, the relationship of propensity to abnormal immune reactions and genetic background should be better discussed. Last but not least, the alterations of gut microbiome are introduced without any attempt to link them to the other findings.

Although the idea of reviewing and juxtaposing pathologies which are differently classified but may share common pathogenetic mechanisms is a very good one, one cannot perceive from the text any suggestion about the interpretation of the data presented or any discussion on the reasons why there are differences and commonalities in these pathologies.

A few typos: ani>>>>anti; fatal>>>fetal